# Genetic Diversity of *Plasmodium vivax* Cysteine-Rich Protective Antigen (PvCyRPA) in Field Isolates from Five Different Areas of the Brazilian Amazon

**DOI:** 10.3390/genes12111657

**Published:** 2021-10-21

**Authors:** Lana Bitencourt Chaves, Glaucia de Oliveira Guimarães, Daiana de Souza Perce-da-Silva, Dalma Maria Banic, Paulo Renato Rivas Totino, Ricardo Luiz Dantas Machado, Rodrigo Nunes Rodrigues-da-Silva, Lilian Rose Pratt-Riccio, Cláudio Tadeu Daniel-Ribeiro, Josué da Costa Lima-Junior

**Affiliations:** 1Laboratory of Immunoparasitology, Oswaldo Cruz Institute, Fiocruz, Rio de Janeiro 21040-900, Brazil; lana.bitencourt@hotmail.com (L.B.C.); goguimaraes.go@gmail.com (G.d.O.G.); 2Laboratory of Basic and Applied Immunology, Arthur Sá Earp Neto University Center, Petrópolis 25680-120, Brazil; daiana_perce@hotmail.com; 3Laboratory of Clinical Immunology, Oswaldo Cruz Institute, Fiocruz, Rio de Janeiro 21040-900, Brazil; dmbanic@gmail.com; 4Laboratory of Malaria Research, Oswaldo Cruz Institute, Fiocruz, Rio de Janeiro 21040-900, Brazil; prtotino@ioc.fiocruz.br (P.R.R.T.); riccio@ioc.fiocruz.br (L.R.P.-R.); malaria@fiocruz.br (C.T.D.-R.); 5Department of Microbiology and Parasitology, Biomedical Institute, Fluminense Federal University, Rio de Janeiro 24210-130, Brazil; ricardomachado@id.uff.br; 6Laboratory of Monoclonal Antibodies Technology, Institute of Immunobiology Technology, Fiocruz, Rio de Janeiro 21040-900, Brazil; rnrsufrrj@gmail.com

**Keywords:** malaria, Brazilian Amazon, *Plasmodium vivax*, PvCyRPA, genetic diversity, prediction, vaccine candidate

## Abstract

The *Plasmodium vivax* Cysteine-Rich Protective Antigen (PvCyRPA) has an important role in erythrocyte invasion and has been considered a target for *vivax* malaria vaccine development. Nonetheless, its genetic diversity remains uncharted in Brazilian malaria-endemic areas. Therefore, we investigated the *pvcyrpa* genetic polymorphism in 98 field isolates from the Brazilian Amazon and its impact on the antigenicity of predicted B-cell epitopes. Genetic diversity parameters, population genetic analysis, neutrality test and the median-joining network were analyzed, and the potential amino acid polymorphism participation in B-cell epitopes was investigated. One synonymous and 26 non-synonymous substitutions defined fifty haplotypes. The nucleotide diversity and Tajima’s D values varied across the coding gene. The exon-1 sequence had greater diversity than those of exon-2. Concerning the prediction analysis, seven sequences were predicted as linear B cell epitopes, the majority contained in conformational epitopes. Moreover, important amino acid polymorphism was detected in regions predicted to contain residues participating in B-cell epitopes. Our data suggest that the *pvcyrpa* gene presents a moderate polymorphism in the studied isolates and such polymorphisms alter amino acid sequences contained in potential B cell epitopes, an important observation considering the antigen potentiality as a vaccine candidate to cover distinct *P. vivax* endemic areas worldwide.

## 1. Introduction

Malaria remains an important public health problem in several countries of tropical and subtropical regions of the world. In 2019, the disease caused an estimated 229 million clinical cases and around 409,000 deaths worldwide [1]. Among the *Plasmodium* species causing malaria in humans, *Plasmodium vivax* is the most widely distributed and prevalent outside of Africa [2]. In Brazil, endemic regions are restricted to the Legal Amazon, a region that currently accounts for the majority (>99%) of the countrywide malaria burden [3] and where *P. vivax* is predominant, with approximately 90% of the reported cases [4]. Several exclusive features of *P. vivax* biology, including the dormant liver stage, make it more resistant than other *Plasmodium* species to malaria elimination [5]. Thus, *P. vivax* presents a difficult obstacle to malaria elimination in endemic countries [6]. Therefore, it is very important to develop new methods and intervention strategies to block or reduce this transmission.

The complex life cycle of the *Plasmodium* includes an erythrocytic phase that is responsible for the clinical symptoms of malaria [7]. In this phase, *P. vivax* preferentially invades reticulocytes [8] in a process that occurs by sequential multiple molecule interactions, with each step mediated by antigens belonging to different protein families present on the merozoite surface and its apical organelles (i.e., micronemes and rhoptries) [9], which interact with a series of specific receptors on the erythrocyte surface to complete the invasion process [10]. The Cysteine-Rich Protective Antigen (CyRPA) is localized in the micronemes and is involved in the invasion process of merozoites into erythrocytes [11]. In *Plasmodium falciparum*, studies with the PfCyRPA protein have already demonstrated its inhibitory role in the invasion of merozoites both in vitro and in vivo, suggesting its potential as a candidate for asexual blood phase vaccine [12,13,14]. However, data demonstrating the potential of *P. vivax* CyRPA (PvCyRPA) as a vaccine candidate are still scarce and conflicting. França and collaborators demonstrated that antibodies against PvCyRPA are strongly related to protection. Interestingly, the protective effect of antibodies directed against PvCyRPA was higher than other proteins classically described as vaccine candidates, such as MSP-1, -3, -9 and AMA-1 [15]. On the other hand, in vitro studies of Ndegwa et al. (2021) showed that polyclonal antibodies raised against full-length PvCyRPA did not affect *P. knowlesi* growth [16].

A vaccine able to produce antibodies that effectively prevent the invasion process after the release of merozoites into the bloodstream may decrease parasite burden, disease symptoms and, indirectly, malaria transmission [17]. However, extensive allelic polymorphism in erythrocyte invasion pathways is known to limit the action of neutralizing antibodies against merozoite candidate vaccine antigens [18]. Malaria parasites have abundant genetic polymorphisms, much of which have evolved to escape host immune responses and thus present a major obstacle to the development of an effective malaria vaccine [19,20]. The genetic diversity and population structure of *P. vivax* for each candidate antigen is an important priority to the understanding of the malaria transmission dynamics [21]. In this scenario, many studies have been proposed to investigate the global diversity of leading vaccine antigens [22], and only one was recently addressed to the PvCyRPA [23], which does not include Brazilian malaria-endemic areas. Therefore, to understand the potential of PvCyRPA in vaccine development, we proposed to identify *pvcyrpa* gene in clinical isolates from different regions of the Brazilian Amazon and to study the potential impacts of the genetic diversity in predicted epitopes through bioinformatics tools.

## 2. Materials and Methods

### 2.1. Study Areas and Blood Sample Collection

Most cases of malaria in Brazil are concentrated in the Amazon Region, an endemic area for the disease [24]. For that reason, the study was carried out in five different regions of the Brazilian Amazon, with a set of 98 *P. vivax*-infected individuals previously described by our group: 31 from Cruzeiro do Sul, 17 from Mâncio Lima, 4 from Guajará, 37 from Manaus and 9 individuals from Oiapoque [25].

All 98 *P. vivax* participants were enrolled according to the following criteria: sought medical assistance for clinical malaria symptoms, presented uncomplicated malaria symptoms, were >18 years of age, and had a positive *P. vivax* malaria diagnosis. Pregnant women and *P. falciparum*-infected individuals were excluded from the study. Thin and thick blood smears were examined for the identification of the malaria parasite by a technician experienced in malaria diagnosis from the Brazilian Malaria Health Services. Thick blood smears from all subjects were stained with Giemsa, and a total of 200 microscopic fields were examined under a 1000-fold magnification. Thin blood smears of the positive samples were examined for species identification. To increase the sensitivity of parasite detection, molecular analyses using specific primers for genus (*Plasmodium* sp.) and species (*P. falciparum* and *P. vivax*) were performed in all the samples as previously described [26]. Donors positive for *P. vivax* and/or *P. falciparum* at the time of blood collection were subsequently treated by the chemotherapeutic regimen recommended by the Brazilian Ministry of Health.

### 2.2. Ethical Considerations

The study protocol was approved by the Research Ethics Committee of each locality, which included obtaining the following patients’ written consents for research use of their blood samples: Cruzeiro do Sul, Mâncio Lima and Guajará were reviewed and approved by the Fundação Oswaldo Cruz Research Ethics Committee, CEP-Fiocruz CAAE 46084015.1.0000.5248. In addition, the protocol of other blood sample collection was approved by the Research Ethics Committee of each locality: Manaus (CEP-Fiocruz): 346–613; Oiapoque (Hospital Municipal do Oiapoque/AP): 68980-000.

### 2.3. Genomic DNA Extraction

The DNA from 98 blood samples was previously extracted using the QIAamp DNA Blood Mini Kit (Qiagen, Hilden, Germany) according to the manufacturer’s instructions and, then, stored at −20 °C until amplification.

### 2.4. Design of Pvcyrpa Specific Primers

The specific primers of *pvcyrpa* gene (1101 bp) were designed using standard gene sequences of *P. vivax* Salvador-1 (Sal-1) strain from GenBank NCBI Reference Sequence: XM_001615090.1 (Gene ID: PVX_090240). All oligonucleotides were designed and checked for specificity by using the Primer-BLAST tool provided by the National Center for Biotechnology Information (http://www.ncbi.nlm.nih.gov/tools/primer-blast/, accessed on 13 August 2019) and the design quality of the oligonucleotides was evaluated by OligoAnalyzer v.3.1 (https://www.idtDNA.com/calc/analyzer/, accessed on 13 August 2019) to avoid homodimers and heterodimers (Table 1). The specific primers were chemically synthesized to perform PCR and DNA sequencing. The *pvcyrpa* gene has a structure consisting of two exons separated by a small well-conserved intron located on chromosome five, encoding for a microneme protein [22,27,28,29]. Consequently, from the extracted genomic DNA, the two exons of the *pvcyrpa* gene were amplified separately using two different primer sets, resulting in two gene lengths.

### 2.5. PCR Amplification of Pvcyrpa Gene

All the *pvcyrpa* genes reported in this study were amplified by conventional PCR using the two pairs of primers designed and described above. PCR reactions of the *pvcyrpa* gene were carried out in 25 μL volume that included 3 μL of DNA, 10 pmol/μL of each primer and the Master Mix kit (Promega, Madison, WI, USA) containing Taq DNA polymerase, PCR buffer and 10 nmol of each deoxynucleotide triphosphate (dNTP, Promega, Madison, WI, USA). The conventional PCR reaction was carried out using a GeneAmp PCR system 9700 (Applied Biosystems, Foster City, CA, USA) and the amplification conditions were as follows: for the first exon, one step at 95 °C for 2 min, 30 cycles at 95 °C for 1 min, 56 °C for 1 min and 72 °C for 1 min, and the last step at 72 °C for 1 min. For the second exon, the temperatures and the number of cycles remained the same, except for the annealing temperature, which was 59 °C for 1 min. The PCR conditions for the amplification of PvCyRPA F1/R1 generated the first *pvcyrpa* gene fragment (600 bp) and the PCR conditions for amplification of PvCyRPA F2/R2 yielded the second *pvcyrpa* gene fragment (466 bp). In all reactions, two negative controls (one without DNA and the other with DNA extracted from in vitro culture of *P. falciparum* PSS1 strain) and positive control (*P. vivax*-infected sample) were used. To confirm the presence of DNA from the in vitro culture of *P. falciparum* and that the lack of amplification was due to the specificity of the primers for *pvcyrpa*, we performed the amplification of the *P. falciparum p126* gene fragment and electrophoresis as previously described [30]. After PCR, ten μL of amplified products were size-fractionated by electrophoresis within 2% agarose gel (Sigma Aldrich, Missouri, USA) in 1× TAE buffer (0.04 M TRIS-acetate, 1 mM EDTA) in the presence of 1× GelRed nucleic acid stain (Biotium, Fremont, CA, USA). PCR products were visualized by ultraviolet (UV) illumination. The sizing of products was performed using a GeneRuler 100 bp Plus DNA Ladder (Thermo Scientific, Waltham, MA, USA). Then, amplicons were purified using the GE Healthcare Lifesciences kit following the manufacturer’s instructions. Afterward, 5–50 ng of DNA was used per sequencing reaction employing the Sanger method, using forward and reverse primers.

### 2.6. DNA Sequencing and Polymorphism Analysis

The specificity of the assay was confirmed by sequencing the PCR products from all positive samples using a Big Dye Terminator Sequencing Kit (Applied Biosystems, Foster City, CA, USA) following the manufacturer’s instructions. The DNA sequencing was carried out on the ABI 3730xl DNA analyzer (Applied Biosystems, Foster City, CA, USA) with the support of Fiocruz Genomic Platform and all the results were analyzed using DNASTAR’s sequence alignment software [31]. Moreover, the sequences were also analyzed in BioEdit sequence alignment editor to better-visualized SNP positions, employing ClustalW multiple sequence alignment and the Sal-1 strain as a reference sequence.

To analyze hypothetical PvCyRPA protein derivatives from *P. vivax* genome data available and Mexico isolates already available, multiple sequence alignments were conducted with Clustal Omega using the MegAlign Pro 15 (Lasergene DNASTAR) program. The following sequences were used: India VII—GenBank: gb|KMZ81773.1; North Korean—GenBank: gb|KNA00954.1; Mauritania I—Genbank: gb|KMZ94334.1; Brazil I—GenBank: gb|KMZ87926.1; Sanger Institute: SCO 66052.1; Sanger Institute: SCO71483.1; Sanger Institute: SGX76259.1; Mexico-Southern Mexican [23]. Additionally, our Brazilian Amazon isolates were compared to P01 strain (PVP01_0532400), a new reference genome for *P. vivax* from an Indonesian clinical isolate [32], and the circular map of protein alignment was generated using the software GenVision v15 (Lasergene DNASTAR).

Multiple alignment Clustal Omega, distance matrix, and the phylogenetic tree were conducted using the MegAlign Pro 15 (Lasergene DNASTAR) program and the circular map of protein alignment was generated using the software GenVision v15 (Lasergene DNASTAR).

### 2.7. Genetic Analysis of the Coding Gene

Genetic diversity of *pvcyrpa* sequences was analyzed using the DnaSP v6 software [33] to estimate within-population diversity based on the genetic diversity parameters as the number of segregation sites (S), the number of haplotypes (h), haplotype diversity (Hd), and nucleotide diversity (π).

Natural selection in *pvcyrpa* was assessed by the Tajima’s D and Z-test. To test the neutral theory of evolution, Tajima’s D values [34] were calculated using the total number of mutations also estimated with DnaSP v.6 software. This test informs about the selection and demographic forces acting on a population. Positive values might be suggestive of positive or balancing selection. This force maintains alleles at balanced frequencies. On the other hand, negative values suggest purifying selection or recent population expansion [34]. The Z-test method was performed with MEGA7 v.6.0; the rates of non-synonymous (dN) to synonymous (dS) substitutions (dN/dS) (1000 bootstrapping replicates) were estimated with Nei and Gojobor’s method [35] and with the Jukes and Cantor correction, in which *p* < 0.05 was considered significant.

Haplotype data also were generated using DnaSP v.6 and the haplotype network was constructed using PopArt v.1.7 with the median-joining algorithm [36] to explore the parasite relationships based on the *pvcyrpa* gene. Mutational steps represent the connections between haplotypes, and empty squares show the non-sampled or extinct haplotypes. The color of the circles represents the geographic origins of each haplotype, while the size of the circle represents the frequency of each haplotype.

### 2.8. Prediction of Linear B-Cell Epitopes

The prediction of linear B-cell epitopes was carried out using the Ellipro algorithm, as well as confirmed by the overlap between the predictions of at least two more algorithms (BCPred, BepiPred, ABCpred and Emini). This software takes a single sequence in FASTA format input and each amino acid receives a prediction score profile of known antigens and incorporates propensity scale methods based on hydrophilicity and secondary structure prediction. For each input sequence, the server outputs a prediction score. The positions of the linear B-cell epitopes are predicted to be located at the residues with the highest scores. In addition, the software ElliPro predicts linear and discontinuous antibody epitopes based on a protein antigen’s 3D structure and accepts two types of input data: protein sequence or structure (PDB format) [37]. This server associates each predicted epitope with a score, defined as a PI (Protrusion Index) value averaged over epitope residues. In the method, residues with larger scores are associated with greater solvent accessibility.

## 3. Results

### 3.1. Molecular Characterization of the Pvcyrpa Gene in the Studied Regions

To identify the gene encoding the PvCyRPA in isolates from Brazilian endemic areas, 98 blood samples from *P. vivax*-infected individuals living in the cities of Cruzeiro do Sul, Mâncio Lima, Guajará, Manaus e Oiapoque had the DNA extracted and subjected to molecular diagnosis by conventional PCR.

The *pvcyrpa* sequence encodes 1101 bp with the two exons sequence and so it was divided into two regions (exon-1 and exon-2). The primer combinations (PvCyRPA_F1/R1 and PvCyRPA_F2/R2) designed to cover the length of the targeted *pvcyrpa* gene resulted in amplification of two fragments in 100% of samples (Figure 1). PvCyRPA F1/R1 primer combination amplified a fragment of 600 bp whereas PvCyRPA F2/R2 primer combination amplified a fragment of 466 bp. Additionally, *P. falciparum* specimens from in vitro culture were tested for quality assurance, resulting in negative PCR amplification of the *pvcyrpa* gene (Figure 1B). Therefore, the 98 samples from individuals infected with *P. vivax* amplified by PCR were subjected to sequencing reactions to screen the possible single nucleotide polymorphisms of the *pvcyrpa* gene. All amplified fragments were sequenced and aligned for sequence analysis.

### 3.2. Genetic Diversity of the Pvcyrpa Gene

Compared to reference sequence Sal-1 (Gene ID: PVX_090240), 27 polymorphic sites were observed, of which one was synonymous and 26 were non-synonymous substitutions. In addition, the isolates presented a frequent deletion of a single codon in position E267 (Table 2). Of these non-synonymous substitutions, two amino acids positions showed two variant alleles each—Q142 (Q142K and Q142R) and D145 (D145G and D145N)—that sometimes were present together in one population. Cruzeiro do Sul and Mâncio Lima presented the two variants existing in position Q142. As well as Cruzeiro do Sul, Mâncio Lima, Manaus and Oiapoque also presented the two variants in position D145, except Guajará. The Cruzeiro do Sul and Mâncio Lima regions shared all the 26 non-synonymous substitutions. Overall, R122K (*N* = 80%; 82%), K131E (*N* = 77%; 79%), D149G (*N* = 62%; 63%), A154D (*N* = 60%; 61%) and E159D (*N* = 66%; 67%) SNPs were the most frequent in Brazilian Amazon isolates, while E86 (*N* = 91%; 93%), Q142 (*N* = 95%; 97%), A187 (*N* = 92%; 94%) and V287 (*N* = 91%; 93%) presented high frequency of similarity to the wild type sequence. *N* (%): Frequency of substitution in each locality (Table 2 and Figure 2).

### 3.3. Population Genetic Analysis

In *P. vivax* isolates from the Brazilian Amazon, the genetic diversity was heterogeneously distributed between the regions coding the two exons, with higher values for exon-1 comparing to exon-2 among localities (Table 3). Using the entire coding gene, it was shown that Mâncio Lima isolates had the highest nucleotide diversity (π) (0.01272 ± 0.00064), while Manaus had the lowest (0.01116 ± 0.00030). Oiapoque had the highest haplotype diversity (Hd) (1.000 ± 0.052), while Guajará had the lowest (0.833 ± 0.222). Some similar differences were detected when analyzing each exon separately. Concerning exon-1, the highest nucleotide diversity was also observed in the Mâncio Lima group (0.01517 ± 0.00095) among all five populations, while Manaus sequences displayed the lowest nucleotide diversity (0.01285 ± 0.00050). Moreover, parasites from Oiapoque presented the highest estimate of haplotype diversity (0.917 ± 0.092), whereas parasites from Guajará showed the lowest Hd (0.667 ± 0.204). In comparison to exon-2, the highest nucleotide diversity was observed in the Mâncio Lima group (0.00938 ± 0.00077). In contrast, Guajará sequences displayed the lowest nucleotide diversity (0.00792 ± 0.00420) in exon-2. Isolates from Oiapoque presented the highest estimate of haplotype diversity (Hd) (0.889 ± 0.091), whereas parasites from Guajará showed the lowest Hd (0.500 ± 0.265).

We performed the Tajima’s D and Z-test to determine whether natural selection was affecting the *pvcyrpa* gene. The Tajima’s D (TjD) test was performed to assess if there is selective pressure on the *pvcyrpa* gene and it was calculated using the total number of mutations. In coding regions, an excess of non-synonymous relative to synonymous changes suggests a clear signal of positive selection. The TjD test showed significant positive values for entire coding in Cruzeiro do Sul (2.24953, * *p* < 0.05) and Manaus (2.75595, ** *p* < 0.01). Similarly, for exon-1 in Cruzeiro do Sul (2.17555, * *p* < 0.05), Guajará (2.24818, * *p* < 0.05) and Manaus (2.60865, ** *p* < 0.01) showed significant positives values, while the other locations showed no significant positive values. In contrast, the TjD test presented a significant positive value at exon-2 only in Manaus (2.42821, * *p* < 0.05). The Z test of selection shows that the exon-1 sequence had positive values in parasites of different origins. These values were higher in all parasites except Guajará (Table 3).

### 3.4. Haplotype Network Analysis

The median-joining haplotype network constructed by PopArt 1.7 using the 98 sequences produced a total of 50 haplotypes with some of which consisting of more than one sequence from the Brazilian Amazon (Figure 3). The haplotype network to explore the parasite relationships based on the *pvcyrpa* gene and comprising mutations at 34 segregating sites. All 50 haplotypes were found closely related. The haplotypes Hap_1 and Hap_11 had high frequency and shared parasites from all five localities. The haplotype Hap_4 (Cruzeiro do Sul, Mâncio Lima and Guajará) and Hap_8 (Cruzeiro do Sul, Mâncio Lima and Manaus) both shared parasites from three localities. Moreover, the haplotypes Hap_2 (Cruzeiro do Sul and Mâncio Lima) and Hap_9 (Cruzeiro do Sul and Manaus), Hap_14 (Cruzeiro do Sul and Mâncio Lima), Hap_23 (Manaus and Oiapoque) shared sequences from 2 localities. The other haplotypes had sequences from only one location.

### 3.5. Comparison of Amino Acid Variations in PvCyRPA among Genome Sequences Available Worldwide

The PvCyRPA amino acid substitutions identified in genome sequences worldwide, including those from Brazilian Amazon, are resumed in Table 4. As observed in the protein sequence alignments, the PvCyRPA coding gene had an excess of non-synonymous mutations, which were more frequent in exon-1 than in exon-2. In addition, we subsequently aligned the protein sequence of these mutant field isolates with other hypothetical CyRPA proteins derivative from *P. vivax* genome data available in the GenBank database and also aligned with the isolate Mexico-Southern Mexican [23]. Among a total of 31 in PvCyRPA protein observed in *P. vivax* sequences worldwide, 26 amino acid substitutions are also present in our isolates. Interestingly, only our isolates showed a new substitution at K150R, being found in the localities of Cruzeiro do Sul, Mâncio Lima, and Guajará. In addition, we can observe a high genetic variability among the isolates of each locality. Curiously, Cruzeiro do Sul and Mâncio Lima present the two variants existing in position Q142 when compared to the genome sequences. Additionally, Cruzeiro do Sul, Mâncio Lima, Manaus and Oiapoque also presented the two variants in position D145, except Guajará. Likewise, some amino acid substitutions of the PvCyRPA protein were found but were rare in only some genomes: L180H in India VII; I63T and Y361H in North Korea; R125T and Q147K in SCO 66052.1 (Sanger Institute) (Table 4). Furthermore, we also compared the consensus sequences of our Brazilian Amazon isolates with reference sequence Sal-1 (PVX_090240) and P01 strain (PVP01_0532400) and generated the circular map of protein alignment using the software GenVision v15 (Lasergene DNASTAR) (Figure 4A). As expected, a significantly high degree of identity was observed across the sequences analyzed, maintaining the mutations found in relation to reference sequence Sal-1. The analysis showed a high identity among our isolates and P01 strain, despite the deletion of 4 amino acids present in the P01 sequence at positions 13–16 (FLFS). According to pairwise distance, the percent identity ranged from 94.5% (P01 vs. GJ) to 99.7% (CZS vs. OIA) (Figure 4A,B).

### 3.6. Polymorphisms and Potential B-Cell Epitopes

We performed in silico prediction for the identification of B cell epitopes present in the PvCyRPA protein using Sal-1 reference and then, seven amino acid sequences were predicted as linear epitopes, (Table 5). All epitopes were initially predicted by the Ellipro algorithm and were confirmed by the overlap between the predictions of at least two more algorithms (BCPred, BepiPred, ABCpred and Emini). The sequences varied from 9 to 19 amino acids and five sequences were inserted in conformational epitopes. The protein appears to have epitopes mainly in the central region and the C terminal region, while the N region terminal does not contain antigenic sequences. Of note, amino acid polymorphisms were detected in regions predicted to contain residues participating in B-cell epitopes (Table 6).

## 4. Discussion

The invasion of the red blood cell by *Plasmodium* merozoites is essential for parasite survival and proliferation. The merozoites have therefore evolved multiple pathways, using various antigenic proteins which aid in the invasion process. Among the merozoite’s invasive proteins are Cysteine-Rich Protective Antigen (CyRPA), which seems to be essential for the parasite’s life cycle during the invasion of erythrocytes and a ligand for reticulocyte invasion [38]. The discovery of the antigen has revamped hope in the search for an effective malaria blood-stage vaccine of *P. vivax*. However, one of the major obstacles to malaria vaccine development is still the low efficiency of proteins used as immunogens in inducing protection, which, in part, can be explained by genetic polymorphisms [39]. It is important to understand the mechanisms of genetic recombination and sequence variation that represent the repertoire of polymorphic malarial surface antigens and that may help in designing vaccines [29,40]. The genetic diversity of these proteins in hyperendemic areas has been described as a limiting factor for the rapid acquisition of protective immunity and, consequently for the development of an effective vaccine. Furthermore, the antigenic polymorphism of *P. vivax* vaccine candidates has been little discussed in unstable transmission areas such as the Brazilian endemic regions [41]. Thus, considering that the epidemiology of malaria in Brazil presents unstable transmission and the knowledge about the genetic polymorphism of *pvcyrpa* remains unknown, we aimed to identify the *pvcyrpa* gene in isolates from different regions of the Brazilian Amazon and to study the potential impacts of the genetic diversity in potential B-cell epitopes.

The identification and analysis of the genetic diversity of the *pvcyrpa* gene in isolates from different geographic regions of the Brazilian Amazon have not been previously studied. Considering the distance among the studied localities and the possible existence of a gene flow of Plasmodium vivax genome among the studied populations, associated with migration of people, could promote the gene flow of the parasite [22] and impact the parasite transmission and dispersion [42,43]. Our first results showed that the *pvcyrpa* gene has high genetic variability in relation to reference sequence Sal-1, presenting 27 polymorphic sites of which one was synonymous and 26 non-synonymous substitutions throughout the sequence. Among these non-synonymous substitutions, two amino acids positions—Q142 (Q142K and Q142R) and D145 (D145G and D145N)—presented one or two variants in our study areas. Overall, R122K (*N* = 80%; 82%), K131E (*N* = 77%; 79%), D149G (*N* = 62%; 63%), A154D (*N* = 60%; 61%) and E159D (*N* = 66%; 67%) mutations were the most frequent in our Brazilian Amazon isolates. The analysis of the *pvcyrpa* gene from the Brazilian Amazon showed that mutations have contributed to generating nucleotide and haplotype diversity. The similarity in the genetic diversity pattern suggests that similar evolutionary forces act on pvcyrpa parasites and that the structural and/or functional properties are consistent. To evade the immune response, genes encoding for antigenic proteins accumulate non-synonymous mutations, which leads to an increase in diversity. In this study, the *pvcyrpa* gene presented non-synonymous mutation accumulation in parasites of different regions, mainly in exon-1. Significant positive values of Tajima’s D indicate balancing selection and population bottlenecks, while negative values suggest the presence of purifying selection or population expansion [34]. Exon-1 had significant positive values in Cruzeiro do Sul, Guajará and Manaus for the Tajima’s Test (TjD) as well as in Manaus at exon-2. The results suggest that polymorphism at pvcyrpa exon-1 is generated by mutation and recombination, and is probably maintained by positive balancing selection pressure, which might represent an evolutionary advantage to the parasite. Exon-1 codes for highly variant domains exposed on the surface of infected red blood cells (RBCs), while exon-2 codes for the more conserved segment [29]. Furthermore, the level of genetic diversity in blood-stage antigens seems to be associated with the degree of exposure to the immune system [29,44].

*P. vivax* biological and genetic characteristics, host immunity, and local vectors may contribute to their different patterns of demographic expansion [45]. Some discrete *P. vivax* lineages can remain stable across time in one of the areas with the highest malaria transmission in the Americas. Relapses can account for some clonal persistence because *P. vivax* strains are repeatedly reintroduced in the population as hypnozoites reactivate [46]. Maybe this context can be to explain why only Mâncio Lima and Oiapoque showed no significant positive values of Tajima’s D. However, genomic epidemiology approaches can help better to reveal the complex distribution of this parasite in the Brazilian Amazon, as well as the relationships with the worldwide genetic diversity.

Moreover, it was possible to identify 50 different haplotypes of *pvcyrpa* gene among the 98 *P. vivax* field isolates from the regions that were analyzed. The haplotype network explores the parasite relationships based on the *pvcyrpa* gene and comprising mutations at 34 segregating sites, and confirmed the extensive genetic diversity observed in *pvcyrpa* sequences. All 50 haplotypes were found to be closely related, with some of which consisting of more than one sequence from the Brazilian Amazon. Regarding the *pvcyrpa* sequences, we observed that haplotypes Hap_1 and Hap_11 had high frequency and shared parasites from all five localities. These findings suggest a global distribution of parasites containing similar *pvcyrpa* genotypes. Additionally, to compare our findings with the PvCyRPA sequences around the world, we observed that it presents a similar genetic profile among the complete genomes of *P. vivax* available on the GenBank Database. Among a total of 31 amino acid substitutions of PvCyRPA protein observed in *P. vivax* sequences worldwide, 26 amino acid substitutions are also present in our isolates.

To develop an effective malaria vaccine that can work in different regions of the world, it is important to include alleles that can induce the immune response and cover the antigenic diversity of *P. vivax* population. Consequently, the existence of the same haplotypes in different malaria-endemic areas and similar genetic profiles worldwide in their results will be important for the rationale of malaria vaccine designs. Moreover, as the immune system could act as selective pressure and the PvCyRPA is emerging as an alternative antigen in vaccine development, we also evaluated the impact of non-synonymous polymorphisms in relation to predicted B-cell epitope sequences.

Amino acid variation was present at peptide regions potentially participating in B-cell epitopes, which supports the idea that this molecule is under selective immune pressure. The seven B-cell potential epitopes have been identified in the PvCyRPA protein, most of which are contained in conformational epitopes, which corroborates its potential as an antibody target. A characteristic of malaria blood-stage antigens is their participation in merozoite invasion and immune evasion. Immunogenicity studies and molecular modeling are essential to determine the importance of PvCyRPA as a vaccine candidate. Targeting molecules important for the *Plasmodium* life cycle might be limited by their antigenic polymorphism or low immunogenicity. Molecular studies provide information about the dynamics of vaccine antigen polymorphisms that can be used to make informed decisions about which parasite alleles to include in vaccine formulations, and to evaluate accurately the efficacy of vaccines tested in malaria-endemic areas [21]. Thus, an effective antigen vaccine should include alleles that induce host immune responses that are sufficiently broad to cover the existing antigenic diversity. Nevertheless, because of the higher genetic diversity of *P. vivax* compared to *P. falciparum*, generating a broad cross-reactive immune response against highly polymorphic asexual stage antigens faces even greater challenges [47].

## 5. Conclusions

In summary, the present study explored the genetic polymorphism of PvCyRPA in field isolates from distinct endemic areas in Brazil, showing a moderate sequence variation, which could influence the potential B-cell epitopes and, consequently, antibody recognition. Despite the observed amino acid changes in the studied population and sequences worldwide, the potential antibody targets did not seem to be significantly affected. However, due to the paucity of information on PvCyRPA genetic diversity and its potential as a vaccine candidate, more studies are necessary to confirm the impact of PvCyRPA polymorphism in naturally acquired immune response and/or vaccine development.

## Figures and Tables

**Figure 1 genes-12-01657-f001:**
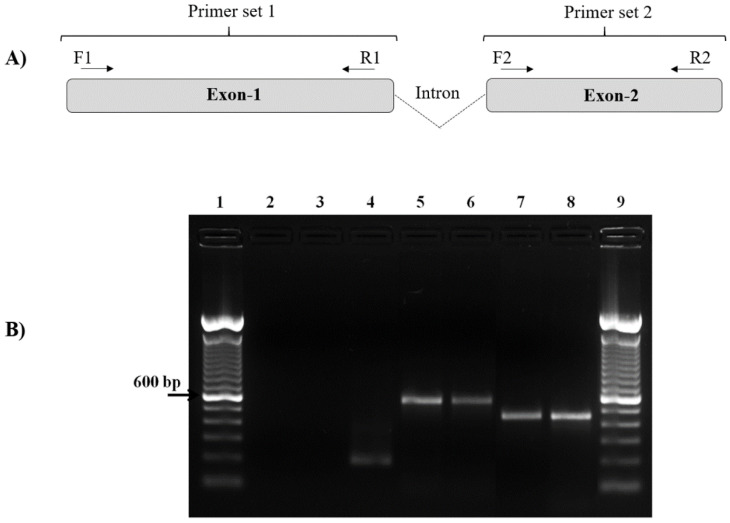
Molecular detection of the *pvcyrpa* gene by PCR. (**A**) Schematic representation of *pvcyrpa* gene, showing the fragments (exons 1 and 2) that were successfully amplified by two different primers sets (PvCyRPA F1/R1 and PvCyRPA F2/R2). (**B**) PCR amplification of the *pvcyrpa* gene. The figure shows agarose gel of *pvcyrpa* fragments amplified using PvCyRPA_F1/R1 primer and PvCyRPA_F2/R2 primer sets, respectively: Lane 1 and 9:100 bp molecular marker; Lane 2: negative control (water); Lane 3: *P. falciparum* in vitro culture (amplification with *pvcyrpa* primers); Lane 4: *P. falciparum* in vitro culture (amplification with *p126* primers); Lane 5 and 7: PCR positive control (*P. vivax* infected sample); Lanes 6 and 8: *P. vivax* samples.

**Figure 2 genes-12-01657-f002:**
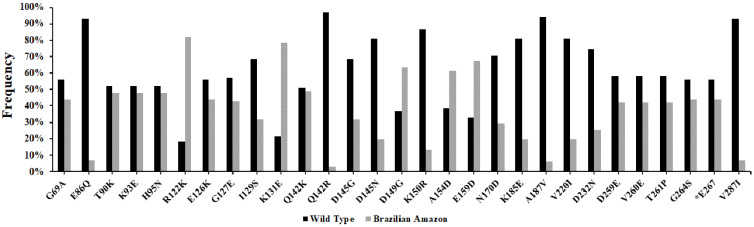
Frequency of substitutions in the PvCyRPA of *P. vivax* isolates from Brazilian Amazon. The two colors (black and gray) represent the two alleles within a population: reference allele and variant allele, respectively. The identification code represents the amino acid in the Sal-1 reference sequence (first character), followed by the position of this residue (number), and the replacing amino acid observed at the same position (last character).

**Figure 3 genes-12-01657-f003:**
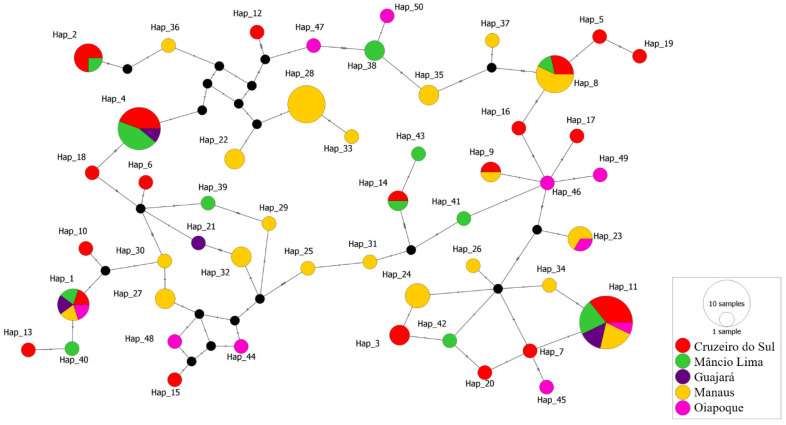
Median-joining network of *pvcyrpa* haplotypes. Each circle represents a unique haplotype and the color of the circles represents the geographic origins of each haplotype, while the size of the circle represents the frequency of each haplotype. Lines separating haplotypes represent mutational steps.

**Figure 4 genes-12-01657-f004:**
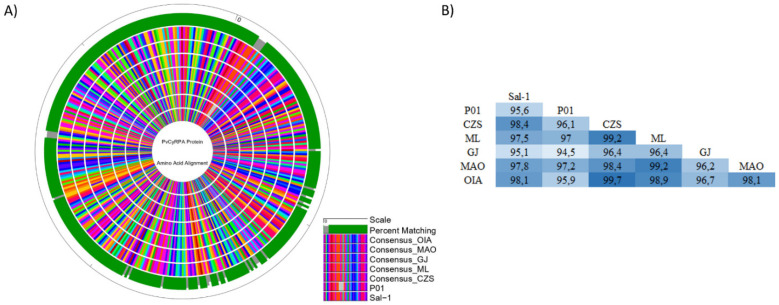
Homology analysis of PvCyRPA protein across isolates. (**A**) Circular alignment of amino acid sequences of PvCyRPA protein with reference sequence Sal-1 and P01 strain. The outer circle shows the amino acid scale. Green and gray bars on the second circle show the percent matching among all sequences used in the analysis. Inner circles show the sequence alignment in which each amino acid was represented by a different color. The legend presents the ordination to outer to inner rings: Scale; Percent Matching; Consensus OIA; Consensus MAO; Consensus GJ; Consensus ML; Consensus CZS; P01 strain; Sal-1 strain. (**B**) The pairwise distance among all sequences studied.

**Table 1 genes-12-01657-t001:** Polymerase chain reaction (PCR) primers used for the amplification of the *pvcyrpa* gene.

	PCR Primers	Direction	Sequence (5′–3′)	Gene Length (bp)
Exon-1	PvCyRPA_F1	Forward primer	TGCAATTTTCCTCTTCTTCTTCC	600
	PvCyRPA_R1	Reverse primer	CCCCATGTTCTTCCCTTGTCTT	
Exon-2	PvCyRPA_F2	Forward primer	CCTGATAAACTACGAAGAGCTCCA	466
	PvCyRPA_R2	Reverse primer	CCTCGTATAGTAAAGCGTGTGT	

Specific primers used for conventional PCR and sequencing of the *pvcyrpa* gene.

**Table 2 genes-12-01657-t002:** Polymorphisms in the PvCyRPA of *P. vivax* isolates from the Brazilian Amazon.

Sal-1 ^a^	Substitutions ^b^	Isolates					
		CZS (31) *N* (%)	ML (17) *N* (%)	GJ (4) *N* (%)	MAO (37) *N* (%)	OIA (9) *N* (%)	Total (98) *N* (%)
G69	G69**A**	12 (39%)	7 (41%)	2 (50%)	19 (51%)	3 (33%)	43 (44%)
E86	E86**Q**	4 (13%)	1 (6%)	0	0	2 (22%)	7 (7%)
T90	T90**K**	14 (45%)	10 (59%)	2 (50%)	19 (51%)	2 (22%)	47 (48%)
K93	K93**E**	14 (45%)	9 (53%)	2 (50%)	19 (51%)	3 (33%)	47 (48%)
H95	H95**N**	14 (45%)	9 (53%)	2 (50%)	19 (51%)	3 (33%)	47 (48%)
R122	R122**K**	24 (77%)	15 (88%)	4 (100%)	30 (81%)	7 (78%)	80 (82%)
E126	E126**K**	14 (45%)	7 (41%)	2 (50%)	17 (46%)	3 (33%)	43 (44%)
G127	G127**E**	14 (45%)	7 (41%)	2 (50%)	17 (46%)	2 (22%)	42 (43%)
I129	I129**S**	9 (29%)	6 (35%)	2 (50%)	11 (30%)	3 (33%)	31 (32%)
K131	K131**E**	23 (74%)	13 (76%)	4 (100%)	30 (81%)	7 (78%)	77 (79%)
Q142	Q142**K**	17 (55%)	8 (47%)	2 (50%)	15 (41%)	6 (67%)	48 (49%)
	Q142**R**	1 (3%)	2 (12%)	0	0	0	3 (3%)
D145	D145**G**	9 (29%)	6 (35%)	2 (50%)	11 (30%)	3 (33%)	31 (32%)
	D145**N**	8 (26%)	2 (12%)	0	6 (16%)	3 (33%)	19 (19%)
D149	D149**G**	23 (74%)	13 (76%)	4 (100%)	16 (43%)	6 (67%)	62 (63%)
K150	K150**R**	6 (19%)	5 (29%)	2 (50%)	0	0	13 (13%)
A154	A154**D**	20 (65%)	11 (65%)	4 (100%)	19 (51%)	6 (67%)	60 (61%)
E159	E159**D**	23 (74%)	14 (82%)	4 (100%)	19 (51%)	6 (67%)	66 (67%)
N170	N170**D**	12 (39%)	8 (47%)	2 (50%)	7 (19%)	0	29 (30%)
K185	K185**E**	10 (32%)	6 (35%)	2 (50%)	0 (0%)	1 (11%)	19 (19%)
A187	A187**V**	2 (6%)	2 (12%)	0	2 (5%)	0	6 (6%)
V220	V220**I**	2 (6%)	2 (12%)	0	13 (35%)	2 (22%)	19 (19%)
D232	D232**N**	9 (29%)	3 (18%)	3 (75%)	7 (19%)	3 (33%)	25 (26%)
D259	D259**E**	12 (39%)	8 (47%)	1 (25%)	18 (49%)	2 (22%)	41 (42%)
V260	V260**E**	12 (39%)	8 (47%)	1 (25%)	18 (49%)	2 (22%)	41 (42%)
T261	T261**P**	12 (39%)	8 (47%)	1 (25%)	18 (49%)	2 (22%)	41 (42%)
G264	G264**S**	14 (45%)	8 (47%)	1 (25%)	18 (49%)	2 (22%)	43 (44%)
E267	***	14 (45%)	8 (47%)	1 (25%)	18 (49%)	2 (22%)	43 (44%)
V287	V287**I**	4 (13%)	2 (12%)	0	1 (3%)	0	7 (7%)

^a^ Reference Sequence Salvador-1; ^b^ Substitutions: The first letter represents the amino acid in the Sal-1 reference sequence and the last in bold the replacing amino acid; Study areas: CZS, Cruzeiro do Sul; ML, Mâncio Lima; GJ, Guajará; MAO, Manaus; OIA, Oiapoque. The number of samples per locality is indicates in parentheses; ***: Deletion of amino acid in position E267; *N* (%): Frequency of substitution in each locality.

**Table 3 genes-12-01657-t003:** Diversity parameters and natural selection for the *pvcyrpa* gene in *P. vivax* isolates from the Brazilian Amazon.

Diversity	Entire Coding	Exon-1	Exon-2
**Cruzeiro do Sul (*n* = 31)**			
No. of segregating sites (S)	32	22	10
No. of haplotypes (h)	20	11	11
Haplotype diversity (Hd)	0.955 ± 0.022	0.875 ± 0.031	0.867 ± 0.038
Nucleotide diversity (π)	0.01248 ± 0.00037	0.01489 ± 0.00050	0.00921 ± 0.00048
Tajima’s test (D)	2.24953 *	2.17555 *	1.96672 ^ns1^
Tajima’s test (Z)	0.774	1.330	0.884
**Mâncio Lima (*n* = 17)**			
No. of segregating sites (S)	32	22	10
No. of haplotypes (h)	11	8	5
Haplotype diversity (Hd)	0.926 ± 0.045	0.846 ± 0.062	0.750 ± 0.088
Nucleotide diversity (π)	0.01272 ± 0.00064	0.01517 ± 0.00095	0.00938 ± 0.00077
Tajima’s test (D)	1.63472 ^ns^	1.59064 ^ns^	1.47204 ^ns^
Tajima’s test (Z)	1.014	1.391	0.743
**Guajará (*n* = 4)**			
No. of segregating sites (S)	20	13	7
No. of haplotypes (h)	3	2	2
Haplotype diversity (Hd)	0.833 ± 0.222	0.667 ± 0.204	0.500 ± 0.265
Nucleotide diversity (π)	0.01168 ± 0.00338	0.01444 ± 0.00442	0.00792 ± 0.00420
Tajima’s test (D)	1.18178 ^ns^	2.24818 *	−0.81734 ^ns^
Tajima’s test (Z)	0.502	0.686	0.440
**Manaus (*n* = 37)**			
No. of segregating sites (S)	27	18	9
No. of haplotypes (h)	19	11	6
Haplotype diversity (Hd)	0.943 ± 0.021	0.886 ± 0.031	0.818 ± 0.024
Nucleotide diversity (π)	0.01116 ± 0.00030	0.01285 ± 0.00050	0.00887 ± 0.00034
Tajima’s test (D)	2.75595 **	2.60865 **	2.42821 *
Tajima’s test (Z)	0.908	1.258	0.685
**Oiapoque (*n* = 9)**			
No. of segregating sites (S)	28	18	10
No. of haplotypes (h)	9	7	6
Haplotype diversity (Hd)	1.000 ± 0.052	0.917 ± 0.092	0.889 ± 0.091
Nucleotide diversity (π)	0.01136 ± 0.00113	0.01361 ± 0.00171	0.00830 ± 0.00233
Tajima’s test (D)	0.74327 ^ns^	1.13901 ^ns^	−0.01607 ^ns^
Tajima’s test (Z)	0.932	1.156	0.402

InDels were not included in the analysis; ^ns^ not significant (*p* > 0.10); ^ns1^ not significant (0.10 > *p* > 0.05); * *p* < 0.05; ** *p* < 0.01.

**Table 4 genes-12-01657-t004:** Comparison of amino acid variations of the PvCyRPA protein between the Brazilian Amazon and genome sequences available worldwide.

	Isolates	Exon-1																							Exon-2							
		Codon Number/Amino Acid Residue																														
		63	69	86	90	93	95	122	125	126	127	129	131	142	145	147	149	150	154	159	170	180	185	187	220	232		260	261	264	287	361
	Sal-1	I	G	E	T	K	H	R	R	E	G	I	K	Q	D	Q	D	K	A	E	N	L	K	A	V	D	D	V	T	G	V	Y

Brazilian Amazon	Cruzeiro do Sul	•	G/A	E/Q	T/K	K/E	H/N	R/K	•	E/K	G/E	I/S	K/E	Q/R/K	D/G/N	•	D/G	K/R	A/D	E/D	N/D	•	K/E	A/V	V/I	D/N	D/E	V/E	T/P	G/S	V/I	•
	Mâncio Lima	•	G/A	E/Q	T/K	K/E	H/N	R/K	•	E/K	G/E	I/S	K/E	Q/R/K	D/G/N	•	D/G	K/R	A/D	E/D	N/D	•	K/E	A/V	V/I	D/N	D/E	V/E	T/P	G/S	V/I	•
	Guajará	•	G/A	•	T/K	K/E	H/N	R/K	•	E/K	G/E	I/S	K/E	Q/K	D/G	•	D/G	K/R	A/D	E/D	N/D	•	K/E	•	•	D/N	D/E	V/E	T/P	G/S	•	•
	Manaus	•	G/A	•	T/K	K/E	H/N	R/K	•	E/K	G/E	I/S	K/E	Q/K	D/G/N	•	D/G	•	A/D	E/D	N/D	•	•	A/V	I	D/N	D/E	V/E	T/P	G/S	V/I	•
	Oiapoque	•	G/A	E/Q	T/K	K/E	H/N	R/K	•	E/K	G/E	I/S	K/E	Q/K	D/G/N	•	D/G	•	A/D	E/D	•	•	K/E	•	V/I	D/N	D/E	V/E	T/P	G/S	•	•

GenBank	gb|KMZ81773.1 IndiaVII	**•**	**•**	**•**	**•**	**•**	**•**	K	**•**	**•**	**•**	**•**	**•**	**•**	**•**	**•**	G	**•**	**•**	**•**	**•**	H	**•**	**•**	**•**	N	**•**	**•**	**•**	**•**	**•**	**•**
	gb|KNA00954.1 North Korean	T	**•**	**•**	K	**•**	**•**	K	**•**	K	E	**•**	E	R	**•**	**•**	G	**•**	D	D	**•**	**•**	**•**	V	I	**•**	**•**	**•**	**•**	**•**	**•**	H
	gb|KMZ94334.1 Mauritania I	**•**	**•**	**•**	**•**	**•**	**•**	K	**•**	**•**		S	E	K	G	**•**	G	**•**	**•**	**•**	**•**	**•**	**•**	**•**	**•**	N	E	E	P	S	**•**	**•**
	gb|KMZ87926.1 Brazil I	**•**	A	**•**	K	E	N	K	**•**	K	E	**•**	E	K	N	**•**	G	**•**	D	D	**•**	**•**	**•**	V	**•**	N	**•**	**•**	**•**	**•**	**•**	**•**

Sanger Institute	SCO 66052.1	**•**	A	**•**	N	**•**	N	K	T	K	E	**•**	E	T	**•**	K	G	**•**	D	D	D	**•**	**•**	V	I	**•**	**•**	**•**	**•**	**•**	I	**•**
	SCO 71483.1	**•**		**•**	K	**•**	**•**	K	**•**	**•**	**•**	**•**	E	R	**•**	**•**	G	**•**	**•**	D	**•**	**•**	**•**	V	**•**	N	**•**	**•**	**•**	**•**	**•**	**•**
	SGX 76259.1	**•**	A	Q	K	E	N	K	**•**	K	E	**•**	E	K	**•**	**•**	**•**	**•**	**•**	D	**•**	**•**	E	P	**•**	**•**	E	E	P	**•**	**•**	**•**

Mexico	Southern Mexican	**•**	A	**•**	**•**	E	N	**•**	**•**	K	E	**•**	**•**	**•**	N	**•**	**•**	**•**	D	D	D	**•**	**•**	**•**	**•**	**•**	E	E	P	S	**•**	**•**

The amino acid variants of the PvCyRPA protein were compared to the reference Sal-1 reference sequence (PVX_090240). • Indicates identical amino acid residues compared to the Sal-1 strain. Codons from 69 to 187 and from 220 to 363 correspond to exon-1 and exon-2, respectively.

**Table 5 genes-12-01657-t005:** B-cell epitope mapping of PvCyRPA (Salvador-1 Strain).

Sequence	Start	End	Lenght	Conformational Epitope	Overlapped Prediction				
					Ellipro	BCPred	ABCpred	BepiPred	Emini
PvCyRPA-B1	81	91	11	Yes	X	X	X	X	-
PvCyRPA-B2	119	129	11	No	X	-	X	X	X
PvCyRPA-B3	134	151	18	No	X	X	X	X	X
PvCyRPA-B4	181	192	12	Yes	X	-	-	X	X
PvCyRPA-B5	241	249	9	Yes	X	X	-	X	-
PvCyRPA-B6	257	272	16	Yes	X	X	-	-	X
PvCyRPA-B7	312	330	19	Yes	X	X	X	-	-

Full-length protein was analyzed for B-cell linear epitopes by the Ellipro algorithm, following confirmation by the overlap between the predictions of the algorithms BCPred, ABCpred, BepiPred, and Emini. (X): confirmed; (-): no confirmed.

**Table 6 genes-12-01657-t006:** Predicted B cell epitopes of PvCyRPA and related polymorphisms.

Epitope	Sequence	Epitope
PvCyRPA_(I81–L91)_	Sal-1	INSTWETQTTL
	E86Q	INSTWQTQTTL
	T90K	INSTWETQTKL
	T90N	INSTWETQTNL
PvCyRPA_(Y119–I129)_	Sal-1	YKQRSKREGTI
	R122K	YKQKSKREGTI
	R125T	YKQRSKTEGTI
	E126K	YKQRSKRKGTI
	G127E	YKQRSKREGTI
	I129S	YKQRSKREGTS
PvCyRPA_(N134–151)_	Sal-1	NSVTGTIYQKEDVQIDKE
	Q142R	NSVTGTIYRKEDVQIDKE
	Q142K	NSVTGTIYKKEDVQIDKE
	Q142T	NSVTGTIYTKEDVQIDKE
	D145G	NSVTGTIYQKEDVQIDKE
	D145N	NSVTGTIYQKENVQIDKE
	Q147K	NSVTGTIYQKEDVKIDKE
	D149G	NSVTGTIYQKEDVQIGKE
	K150R	NSVTGTIYQKEDVQIDRE
PvCyRPA_(S181–F192)_	Sal-1	SYEYKTANKDNF
	K185E	SYEYETANKDNF
	A187V	SYEYKTVNKDNF
	A187P	SYEYKTPNKDNF
PvCyRPA_(R241–R249)_	Sal-1	RISTNNTAR
PvCyRPA_(T257–C272)_	Sal-1	TLDVTNEGKKEYKFKC
	D259E	TLEVTNEGKKEYKFKC
	V260E	TLDETNEGKKEYKFKC
	T261P	TLDVPNEGKKEYKFKC
	G264S	TLDVTNESKKEYKFKC
PvCyRPA_(T312–G330)_	Sal-1	TEQNAIVVKPKVQNDDLNG

Sequences containing polymorphism inserted on predicted epitopes and marked in red letters.

## Data Availability

The data supporting the conclusions of this article are provided within the article.

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
