# Peer review of "Genetic Diversity of Plasmodium vivax Cysteine-Rich Protective Antigen (PvCyRPA) in Field Isolates from Five Different Areas of the Brazilian Amazon"

_genes, 2021, doi:10.3390/genes12111657_

Round 1
Reviewer 1 Report
The manuscript “Genetic diversity of Plasmodium vivax Cysteine-Rich Protective Antigen (PvCyRPA) in field isolates from five different areas of the Brazilian Amazon” by Lana Bitencourt Chaves et al. reports an assessment of genetic diversity across Brazilian populations of a Plasmodium antigen sequence by PCR. Overall, the study is of good quality with robust techniques and analyses. PCR design is quite elaborate with adequate controls. The description of the applied methods is very detailed and overall excellent. Drawn conclusions are supported by presented results. The manuscript would benefit from an increase in consistency of sentences, especially in the results section, and a general proofreading of the grammar. I recommend the following revisions:
Major revisions:
- Please revise the English of the manuscript thoroughly. I have exemplified some sentences in the minor revisions part.
- Did the authors check polymorphic loci in the respective chromatograms (with FinchTV or similar)? Erroneous automated base calling should be ruled out as a source of variant positions.
- Figure 1: I would prefer having the original agarose gel picture. While the composition presented here looks very nice, and understandably displays the whole PCR assay design, original photos allow the reader to draw their own conclusions about band’s position relative to ladder etc. At least a representative should be made available as a supplement.
- Figure 2: If I understand correctly, the black and grey bars sum up to 100% for each position. Why not present a stacked bar chart then instead? Also, the legend names should be revised to make more understandable that the two colors represent the two alleles within a certain group of individuals (e.g., “reference allele” and “variant allele”).
- Paragraph 3.3. Population genetic analysis: Two things are mingled here. The authors talk about “an excess of non-synonymous relative to synonymous changes” in the context of Tajima's D. Tajima's D, however, is a test statistic to infer drift and selection from rates of segregating sites by pairwise haplotype comparisons, irrespective of their effect on resulting protein sequences. The correct population genetic statistic for the comparison of synonymous to non-synonymous substitution rates is dN/dS, which takes counts of synonymous and non-synonymous sites and synonymous and non-synonymous substitutions for estimating the direction of selection on protein-coding genes. The authors should disentangle this and ideally give results for both tests. Importantly, for Tajima's D values of < 0 are indicative of positive selection, whereas for dN/dS values of > 1 are indicative of positive selection. By the way, the statement “27 polymorphic sites of which one was synonymous and 26 were non-synonymous substitutions” (238 ff.) is indicative of very strong positive selection. This actually needs some serious discussion (partly already included).
Minor revisions:
- 48: “more resistant” … More requires something to compare with. More than any other Plasmodium species?
- 48 f.: “is predicted to present the difficult obstacle” … please revise sentence
- 126: micronemes are not familiar to everyone; maybe give short information somewhere in the article
- 169 f.: “multiple sequence alignment Clustal Omega was done” > “multiple sequence alignments were done with Clustal Omega”?
- 187: “the total number or mutations” > “the total number of mutations”?
- 201: “These software takes” > “This software takes”
- 207 f.: “and its accepts as an input protein structure in PDB format” … please revise sentence
- 214: “samples from P. vivax-infected individuals living in the cities of” > rephrase if you aren’t sure whether all of them lived in the city where they’ve been tested
- Paragraph 3.2. Genetic diversity of the pvcyrpa gene: needs significant syntax revision to increase comprehensibility.
- Table 2: This might go into Supplements.
- The recurring phrasing that “2 amino acids positions – Q142 (Q142K and Q142R) and D145 (D145G and D145N) – presented one or two variants” should be re-phrased since they are unclear. I recommend something like “two amino acid positions showed two variant alleles each (Q142K and Q142R; D145G and D145N), that sometimes were present together in one population”.
Author Response
Dear Editor and Reviewers:
Please see the attachment.

Reviewer 2 Report
The manuscript by Bitencourt Chaves et al., is a well-written and a detailed investigation of the genetic diversity of one gene, coding for CyRPA in Plasmodium vivax populations of the Brazilian Amazon. Overall, the data is well presented and the hypothesis and conclusions well described. If PvCyRPA was a suitable vaccine candidate for P. vivax malaria than the information in this manuscript would be an important and valued contribution, necessary for the most effective vaccine design. However, the importance of PvCyRPA in the blood stage life cycle of P. vivax is not certain and the accessibility by antibodies equally is unclear in this species.
Major corrections:
I would encourage the authors to include this uncertainty of the role of PvCyRPA in the blood stage life cycle of P. vivax in the introduction. Whereas Franca et al. demonstrated by using population studies that antibodies to PvCyRPA are linked with a strong protection from disease, Ndegwa et al (2021) showed that polyclonal antibodies raised against full-length PvCyRPA had no effect on parasite growth of the closely related P. knowlesi parasite in vitro. If one was to suggest that P. knowlesi is a good in vitro blood stage model for P. vivax then the protective role of antibodies against PvCyRPA during red cell invasion is in doubt.
In lines 365-366, in the discussion, the authors state the PvCyRPA seems to be essential for the parasite life cycle and act during invasion of erythrocytes as a ligand for reticulocyte invasion quoting Knuepfer et al (2019). In that paper however the role of PkCyRPA was studied, not PvCyRPA. PkCyRPA was found to be essential for the P. knowlesi life cycle in vitro, studying invasion of human normocytes. It is possible that PvCyRPA plays a very similar role to PkCyRPA but this was not demonstrated in the study by Knuepfer et al (2019), which was referenced.
Minor corrections:
- Fig 1b: could the authors please include the original data in the supplementary and indicate the lanes used to assemble Fig1b.
- Table 3: a very useful data summary, but could the authors please separate the data associated for each geographic region more visibly (suggestion: lines or boxes) to help readers associate the data with the correct location more easily?
- Fig 4a: the legend key needs to be larger and the colour labelling makes it unclear which circle represents which strain/consensus sequence. Suggestion: in key refer to outer to inner rings or number these 1-8.
Author Response

(The authors gave the same response as above.)
